# TRPV6 Channel Is Involved in Pancreatic Ductal Adenocarcinoma Aggressiveness and Resistance to Chemotherapeutics

**DOI:** 10.3390/cancers15245769

**Published:** 2023-12-08

**Authors:** Gonçalo Mesquita, Aurélien Haustrate, Adriana Mihalache, Benjamin Soret, Clément Cordier, Emilie Desruelles, Erika Duval, Zoltan Pethö, Natalia Prevarskaya, Albrecht Schwab, V’yacheslav Lehen’kyi

**Affiliations:** 1Laboratory of Cell Physiology, INSERM U1003, Laboratory of Excellence Ion Channel Science and Therapeutics, Department of Biology, Faculty of Science and Technologies, University of Lille, 59650 Villeneuve d’Ascq, Franceemilie.desruelles@univ-lille.fr (E.D.); natacha.prevarskaya@univ-lille.fr (N.P.); 2Institute of Physiology II, University of Muenster, Robert-Koch-Str. 27b, 48149 Muenster, Germany; pethoe@uni-muenster.de; 3Service d’Anatomie et de Cytologie Pathologiques, Groupement des Hôpitaux de l’Institut Catholique de Lille (GHICL), 59000 Lille, France; mihalache.adriana@ghicl.net (A.M.); duval.erika@ghicl.net (E.D.)

**Keywords:** PDAC, TRPV6, calcium, therapy, resistance

## Abstract

**Simple Summary:**

Pancreatic ductal adenocarcinoma (PDAC) is a highly aggressive cancer with limited treatments and poor prognosis. TRPV6, a calcium-permeable channel overexpressed in cancers, holds potential as an influencer of cancer cell behavior. This study investigates TRPV6 expression in PDAC, analyzing 46 patient tissue samples of varying stages and grades. We manipulated TRPV6 expression (knockdown, overexpression) in the human PDAC cell lines Panc-1 and Capan-1. We then revealed the impact of TRPV6 expression on Ca^2+^ influx, proliferation, apoptosis, migration, chemoresistance, and tumor growth both in vitro and in vivo. Notably, TRPV6 expression correlated with tumor stage and grade, relating it to PDAC proliferation. Knockdown decreased Ca^2+^ influx. This was paralleled by reduced proliferation, enhanced apoptosis, and sensitization of cells to chemotherapeutic drugs. Conversely, TRPV6 overexpression yielded opposing effects. Strikingly, both knockdown and overexpression curtailed tumor formation in vivo. These findings underscore an intricate role of TRPV6 channels in PDAC progression, highlighting its potential as a therapeutic target.

**Abstract:**

Pancreatic ductal adenocarcinoma (PDAC) stands as a highly aggressive and lethal cancer, characterized by a grim prognosis and scarce treatment alternatives. Within this context, TRPV6, a calcium-permeable channel, emerges as a noteworthy candidate due to its overexpression in various cancers, capable of influencing the cell behavior in different cancer entities. Nonetheless, the exact expression pattern and functional significance of TRPV6 in the context of PDAC remains enigmatic. This study scrutinizes the expression of TRPV6 in tissue specimens obtained from 46 PDAC patients across distinct stages and grades. We manipulated TRPV6 expression (knockdown, overexpression) in the human PDAC cell lines Panc-1 and Capan-1. Subsequently, we analyzed its impact on multiple facets, encompassing Ca^2+^ influx, proliferation, apoptosis, migration, chemoresistance, and tumor growth, both in vitro and in vivo. Notably, the data indicate a direct correlation between TRPV6 expression levels, tumor stage, and grade, establishing a link between TRPV6 and PDAC proliferation in tissue samples. Decreasing TRPV6 expression via knockdown hampered Ca^2+^ influx, resulting in diminished proliferation and viability in both cell lines, and cell cycle progression in Panc-1. The knockdown simultaneously led to an increase in apoptotic rates and increased the susceptibility of cells to 5-FU and gemcitabine treatments. Moreover, it accelerated migration and promoted collective movement among Panc-1 cells. Conversely, TRPV6 overexpression yielded opposing outcomes in terms of proliferation in Panc-1 and Capan-1, and the migration of Panc-1 cells. Intriguingly, both TRPV6 knockdown and overexpression diminished the process of tumor formation in vivo. This intricate interplay suggests that PDAC aggressiveness relies on a fine-tuned TRPV6 expression, raising its profile as a putative therapeutic target.

## 1. Introduction

Pancreatic ductal adenocarcinoma (PDAC) is the most common type of pancreatic cancer, accounting for about 90% of all cases [1]. It is one of the most lethal malignancies, with a 5-year survival rate of only approximately 10% [2]. The poor prognosis of PDAC is mainly due to its late diagnosis, aggressive invasion, metastasis and resistance to conventional therapies [3]. Therefore, there is an urgent need to develop alternative therapeutic concepts for this disease.

TRPV6 is a Ca^2+^-permeable channel that belongs to the transient receptor potential (TRP) family of ion channels [4,5]. It plays an important physiological role in Ca^2+^ reabsorbing intestinal and renal epithelia. However, TRPV6 is also aberrantly expressed in various cancers, such as prostate, breast, colon, and ovarian cancers [4,6]. In ovarian cancer, TRPV6 has increased expression in all stages when compared to normal tissue [7]. Furthermore, the inhibition of TRPV6 channels in an SKOV-3 xenograft model in mice led to reduced tumor formation. TRPV6 overexpression was also present in breast cancer, where it correlated with invasive areas of the tumor [8]. In the prostate cancer cell line LNCaP, TRPV6 inhibition can lead to lower levels of proliferation and increased apoptosis [9]. Translocation of TRPV6 to the plasma membrane and increased aggressiveness was also observed in LNCaP, indicating a clear role for TRPV6 in prostate cancer [10]. Overall, TRPV6 channels promote cancer cell proliferation, survival, migration, and invasion by modulating intracellular Ca^2+^ signaling and downstream pathways [11,12,13,14]. 

The expression and function of TRPV6 in PDAC are still unclear. A previous study indicated that the overexpression of TRPV6 in PDAC tissues and cell lines correlated with poor survival and chemoresistance [15]. Another study showed that TRPV6 has a reduced expression in micro-dissected PDAC samples [16]. These contradictory findings may reflect the heterogeneity and complexity of PDAC. Furthermore, it is known that TRPV6 impacts non-alcoholic early onset pancreatitis, a risk factor for the development of PDAC [17,18]. This was observed in several cohorts and appears to be correlated with the Ca^2+^ channel activity of TRPV6 [19]. In this study, we aimed to investigate how TRPV6 impacts PDAC aggressiveness in tumor tissue, in PDAC cell line Panc-1, and in vivo. To achieve that, TRPV6 expression was quantified in tumoral and peri-tumoral tissues from PDAC patients. Stable clones from Panc-1 cells were created, with a knockdown or an overexpression of TRPV6. These cells were evaluated for proliferation, viability or motility. These stable clones were also grafted into mice to assess the role of TRPV6 channels in PDAC tumor aggressiveness in vivo.

## 2. Materials and Methods

### 2.1. Human Tissue Data

Tumor and peri-tumoral tissues were obtained from patients with PDAC that was histopathologically confirmed by the medical staff of GHICL (Groupement des Hôpitaux de l’Institut Catholique de Lille, Lille, France) (TNM classification, 8th edition of the UICC 2016). The number of subjects is 46 and tissues were collected between January of 2014 and December of 2019 (RNIPH-2022-26).

For immunohistochemistry, four micrometer-thick conventional whole tissue sections were cut from formalin-fixed and paraffin-embedded (FFPE) tissue blocks. Immunohistochemical staining was performed using a BenchMark ULTRA system (Ventana Medical Systems Inc., Oro Valley, USA) and a revelation kit (ultraView DAB Detection Kit (Roche Diagnostics, Indianapolis, USA)). The sections were immunolabeled with anti-TRPV6 Clone Mab 82 antibody [20] or anti-Ki-67 by using heat-induced epitope retrieval (HIER) in a citrate-based buffer (CCS 2) at 91 °C for 8 min followed by primary antibody incubation for 32 min. TRPV6 expression was analyzed semi-quantitatively with a scoring system ranging from 0 (no staining) to 3 (very intense staining). The staining intensity is based on the H-score, which is the result of the multiplication of the percentage of cells by the score given [21].

### 2.2. Cell Culture

Panc-1 and Capan-1 cells were incubated in a humidified atmosphere at 37 °C and 5% CO_2_. The highly aggressive pancreatic adenocarcinoma cells were cultured in RPMI medium (Roswell Park Memorial Institute medium, Sigma-Aldrich, St. Louis, USA) supplemented with 10% fetal bovine serum (FBS Superior, Berlin, Germany) and 500 μM of geneticin (G418) (ThermoFischer Scientific, Waltham, USA).

### 2.3. Cell Transfection

For transfection, the cells were seeded in 6-well plates at a density of 1 × 10^5^ cells per well and incubated overnight. The next day, the cells were transfected with one of the following plasmids: vEF1ap-5′UTR-TRPV6_CMVp-mCherry-Vektor, vEF1ap-5′UTR_CMVp-mCherry or pSingle-tTS-shRNA-Vektor (Clontech Laboratories, Mountain View, USA). The latter vector had Luciferase or TRPV6 as the shRNA sequence. The plasmids were diluted in Opti-MEM Reduced Serum Medium and mixed with Lipofectamine 3000 Transfection Reagent (Thermo Fisher Scientific) according to the manufacturer’s instructions. The transfection complexes were added to the cells and incubated for 24 h. The medium was then replaced with fresh RPMI with 500 μM of geneticin (G418) in order to select transfected cells. After 2 weeks, a small group of cells (3–4 cells) was selected, and a new culture was prepared for each mixed clone. All the experiments were performed with the stable mixed clones.

### 2.4. RNA Sequencing

The cells were plated in 60 mm dishes at 75% confluence and the total RNA was extracted and purified from the cells using the NucleoSpin^®^ RNA Plus kit (Machery-Nagel, Strasbourg, France) according to the manufacturer’s recommendations. Each RNA sample was validated for RNA integrity using a 18S/28S ratio. In total, 1 µg of total RNA from each sample was used for library preparation. Library preparation was realized following the manufacturer’s recommendations (Illumina Stranded mRNA Prep, Illumina, San Diego, USA). The final samples’ pooled library prep was sequenced on ILLUMINA Novaseq 6000 with SP-200 cartridge (2 × 800 Millions of 100 bases reads), corresponding to 2 × 26 Millions of reads per sample after demultiplexing. This work benefited from equipment and services from the iGenSeq core facility, at ICM (Paris, France). The results were then evaluated as the fold change in the controls.

### 2.5. RT-PCR

cDNA amplification conditions included the initial denaturation step of 7 min at 95 °C and 36 cycles of 30 s at 95 °C, 30 s at 60 °C, 30 s at 72 °C, and finally 7 min at 72 °C. Quantitative real-time PCR of TRPV6 and hypoxanthine-phosphoribosytransferase (HPRT) mRNA transcripts were carried out using MESA GREEN qPCR MasterMix Plus for SYBR Assay (Eurogentec) on the Biorad CFX96 Real-Time PCR Detection System. The sequences of primers are as follows: *TRPV6*: Forward GCCTTCTATATCATCTTCC; Backward GGTGATGCTGTACATGAAGG; HPRT: Forward GGCGTCGTGATTAGTGATGAT; Backward CGAGCAAGACGTTCAGTCCT. The *HPRT* gene used has a housekeeping gene. To quantify the results, the comparative threshold cycle method ∆∆Ct and CFX Manager Software v2.0 were used.

### 2.6. Western Blot

Western blot protocol was followed like in Haustrate et al. [20]. Briefly, the cells were treated with an ice-cold lysis buffer containing the following: 10 mM Tris-HCl, pH 7.4, 150 mM NaCl, 10 mM MgCl_2_, 1 mM PMSF, 1% Nonidet P-40, and protease inhibitor cocktail from Sigma-Aldrich. After centrifugation, the lysates were mixed with a sample buffer containing 125 mM Tris-HCl pH 6.8, 4% SDS, 5% β-mercaptoethanol, 20% glycerol, and 0.01% bromophenol blue, and boiled for 5 min at 95 °C. The total protein samples were subjected to 8% SDS-PAGE and transferred to a nitrocellulose membrane via semi-dry Western blotting (Bio-Rad Laboratories, Hercules, CA, USA). The membrane was blocked overnight in 5% milk containing TNT buffer (Tris-HCl, pH 7.5, 140 mM NaCl, and 0.05% Tween 20) and then probed using specific mouse monoclonal anti-TRPV6 antibodies (all at 1/500 dilution from the initial concentration of 0.5 µg/µL) and mouse monoclonal anti-β-actin (Lab Vision Co., Fremont, CA, USA, 1/1000) antibodies. Mouse monoclonal anti-mouse secondary antibodies (Chemicon International; Temecula, CA, USA, 1/200) were used. The bands on the membrane were visualized using an enhanced chemiluminescence method (Pierce Biotechnologies Inc., Escondido, CA, USA). Densitometric analysis was performed using a Bio-Rad image acquisition system (Bio-Rad Laboratories).

### 2.7. Mn^2+^ Quench Experiments

Panc-1 cells were stained with the Ca^2+^-sensitive dye Fura2, coupled with the AM ester to be able to permeate the cell (3 µM, Invitrogen, Waltham, USA) in Ringer’s solution (NaCl 140 mM, CaCl_2_ 1.2 mM, MgCl_2_ 0.8 mM, KCl 5.4 mM) with Hepes 20 mM for 30 min in a heating cabinet. The cells were then placed on the stage of a Axiovert 200 fluorescence microscope (Zeiss, Oberkochen, Germany) and continuously superfused with prewarmed (37 °C) HEPES-buffered Ringer’s solution. During the experiment, a 3 min control period (Ringer’s solution without Ca^2+^) preceded the application of the test solution (Ringer’s solution with 400 μM MnCl_2_) for 3 min. Mn^2+^ has a higher binding affinity to Fura2 than Ca^2+^ and quenches its fluorescence. Images were taken every other 10 s. Fura2-loaded cells were excited at the isosbestic wavelength of 357 nm [22]. The mean cellular fluorescence intensity emitted at 510 nm was measured following background subtraction. Then, the slope of the fluorescence declines over 30 s for each time point during the Mn^2+^ influx was calculated, and the slope of the control period was subtracted. The steepest slope over 30 s was determined and taken as a surrogate of Ca^2+^ influx.

### 2.8. Cell Count

Cell proliferation was assessed through cell count. Briefly, the cells were plated in a T25 flask at a density of 30,000 cells per flask. Images of proliferating Panc-1 cells were acquired every other 30 min for 48 h, starting 2 h after plating. Cell proliferation was also assessed through mitosis count. This allowed us to exclude cells migrating from other areas. Images were obtained in a Zeiss Axio1040 microscope (Zeiss, Germany) with controlled temperature. The pictures were taken using MikroCamLab II 7.3.1.8 software (Bresser, Germany).

### 2.9. Cell Cycle Assay

Cell cycle analysis was performed by means of flow cytometry of cell populations cultured in triplicate in 30 cm^2^ dishes, as described in the work of Lehen’kyi et al. 2011 [23]. Briefly, Panc-1 cells were washed with phosphate-buffered saline (PBS) and then trypsinized. After this procedure, they were fixed in cold methanol for 30 min at +4 °C. After this step, the cells were centrifuged and with PBS at 4 °C, resuspended in 100 µL PBS, treated with 100 µL RNAse A (1 mg/mL, Sigma-Aldrich), and stained with propidium iodide (PI, Sigma-Aldrich) at a final concentration of 50 µg/mL. The stained cells were stored at 4 °C in the dark and analyzed within 2 h. PI fluorescence was measured with a BD FACSCalibur (BD Biosciences, Franklin Lakes, NJ, USA). Data were acquired for 10,000 events, and red fluorescence was measured using a fluorescence detector 3 (FL3). The data were stored and analyzed using CellQuest software to assess the cell cycle distribution patterns between the different phases: subG1 (apoptotic), G0/G1, S, and G2/M phases.

### 2.10. Cell Viability Analysis

For these experiments, the cells were seeded in 96-well plates at a density of 3 × 10^4^ cells per well. The cells were maintained under normal cell culture conditions in 100 µL medium until the experiment, which occurred every day from day 0 (3 h after plating) up until day 4. To perform the experiment, 100 μL of CellTiter (Promega) was added to each well, followed by 2 min of agitation. The analysis was made 10 min after the addition of CellTiter. Luminescence in the wells was measured by using the TriStar LB 94 Luminometer (Berthold technologies, Bad Wildbad, Germany) controlled via the MikroWin 2000 Software 4.0 software.

### 2.11. MTS Assay

Cell viability was measured using the CellTiter 96 Aqueous One Solution cell proliferation assay (Promega, Madison, WI, USA) on the basis of the cellular conversion of the colorimetric reagent MTS [3,4-(5-dimethylthiazol-2-yl)-5-(3-carboxymethoxyphenyl)-2-(4-sulfophenyl)-2H-tetrazolium salt] into soluble formazan by dehydrogenase enzymes found only in metabolically active cells. Following each treatment, 20 μL of dye solution was added into each well in a 96-well plate and incubated for 2 h. Subsequently, absorbance was recorded at 490 nm wavelength using an ELISA plate reader (Molecular Devices, Sunnyvale, CA, USA).

### 2.12. Annexin V Staining

Annexin V staining was performed with the Annexin V-FITC Apoptosis Detection Kit (ab14085, Abcam, UK) following the manufacturer’s instructions. The cells were incubated under normal cell culture conditions at a density of 5 × 10^5^ cells/mL in glass bottom dishes pre-coated with 0.01% poly-l-lysin. After 3 h, the cells were treated with gemcitabine, cisplatin, 5-fluorouracil or DMSO. After 24 h, the cells were stained with Annexin V-FITC for 15 min. Cell death was analyzed through the intensity of fluorescence in 10 cells on each area. A total of 7 areas were counted for each experiment. Fluorescence microscopy was performed using a Nikon A1 confocal microscope with a filter set for FITC, with a 20× magnification. The cells that bound Annexin V-FITC showed green staining on the plasma membrane. The images were acquired and processed using NIS-Elements 4.0 software from Nikon and the mean fluorescence intensity of 10 cells per visual field were measured.

### 2.13. Wound-Healing Assay

The cells were cultured in RPMI supplemented with 10% FBS in glass bottom dishes, pre-coated with 0.01% poly-l-lysin, with a spacer until they reached confluence. The spacer was removed to create a cell-free gap and replaced the medium with fresh RPMI containing 20 mM HEPES, and the cells were incubated in a heating cabinet at 37 °C for 2 h. The dishes were then transferred to a live cell imaging system (Zeiss Axio1040 microscope, Zeiss) and images of the wound area were captured every other 30 min for 20 h. The temperature was maintained at 37 °C throughout the experiment. Images were analyzed using NIS-elements software and the wound closure rate was assessed in μm^2^ by determining the area of the cells migrating into the wound during the course of the experiment.

### 2.14. In Vivo Experiments

Studies involving animals, including housing and care, method of euthanasia, and experimental protocol (APAFIS #22266), were conducted in accordance with the animal ethical committee (CEEA75) in the animal house (C59-00913) of the University of Lille. Panc-1 cells (2 × 10^6^ cells per mouse) were injected s.c. in 50% (vol:vol) matrigel (BD Biosciences) into 6- to 8-week-old male Swiss nude mice (Charles Rivers Laboratories, Wilmington, USA). Each stable clone was injected into 10 mice. Tumor growth was recorded once per week using a sliding caliper. The mice were sacrificed after 10 weeks. Humane endpoints were prepared for as soon as either a critical tumor size of 2500 mm^3^ or a deviation of more than 10% of the body weight is reached.

### 2.15. Statistical Analysis

Experimental data are shown as mean  ±  SEM. The statistical analysis was performed in GraphPad Prism 8.0. ANOVA followed by the Holm–Sidak test, with multiple comparisons between the different clones in different time points or with different concentrations. *T*-test was used for differences between two groups with one variable. In the graphs, (*) denotes statistically significant differences with *p* < 0.05.

## 3. Results

### 3.1. TRPV6 Expression in PDAC Tumor Tissues Correlates with Tumor Stage, Differentiation and Proliferation

To investigate the role of TRPV6 in pancreatic ductal adenocarcinoma (PDAC), the expression of TRPV6 was analyzed in tissue samples from 46 patients with PDAC in different stages. The staining intensity of TRPV6 was compared in tumor tissues against peri-tumoral tissues (as the control) and graded according to the scoring system ranging from 0 (no staining) to 3 (very intense staining) (Figure 1A). For quantitative analysis, we used the H-score, meaning that the staining intensity is the result of the multiplication of the percentage of cells by a certain score given. Based on the TMN, the tumor tissues were separated into lower disease progression (T1/T2) or higher disease progression (T3/T4) (Figure 1B). Histopathological features were used to classify the tumor tissues as highly differentiated or poorly differentiated (Figure 1C). We also investigated the relationship between TRPV6 expression and cell proliferation in PDAC tumor tissues using the proliferation marker Ki-67. The tumor tissues were grouped into four categories based on the percentage of Ki-67 positive cells: 0–5% (lower proliferation), 5–25%, 25–50%, and >50% (higher proliferation). TRPV6 staining intensity was then analyzed as a function of Ki-67 expression (Figure 1C).

TRPV6 expression was found increased with the tumor stage, from the early stages (stages I and II) to the later ones (stages III and IV) (Figure 1D). TRPV6 staining intensity rises by 67.9 ± 23.1% for tumor tissues and 18.3 ± 10.5% for peri-tumoral tissues between the early and late stages. Furthermore, we found that poorly differentiated tumor tissues express more TRPV6 than highly differentiated ones (Figure 1E). Poorly differentiated tissues had a TRPV6 staining intensity of 126.9 ± 16.2. Well-differentiated tissues only had a staining intensity of 81.7 ± 10.1, indicating a 32% decrease in staining compared to poorly differentiated tissues.

A tendency can be observed for an increase in TRPV6 expression with increasing Ki-67 positivity (Figure 1F). It reaches significance for the group with >50% Ki-67 positive cells. The mean TRPV6 staining intensity for each group was as follows: 61.3 ± 9.7 for 0–5%, 72.2 ± 16.5 for 5–25%, 93.7 ± 16.2 for 25–50%, and 230 ± 30 for the group with >50% of Ki-67 positive cells. This indicates that TRPV6 expression is elevated in highly proliferative PDAC tumor cells.

Taken together, our results suggest that TRPV6 expression appears to correlate with PDAC aggressiveness. The following experiments are aimed at revealing how TRPV6 channels may contribute to PDAC aggressiveness.

### 3.2. Characterization of Panc-1 Cells with Altered TRPV6 Expression

In order to investigate the role of TRPV6 channels in mechanistic studies, two groups of stable clones of both Panc-1 and Capan-1 cells were created: one with a knockdown of TRPV6 expression (Panc-1-shTRPV6 or Capan-1-shTRPV6) and one with an overexpression of TRPV6 (Panc-1-mCherry-TRPV6 or Capan-1-mCherry-TRPV6). Each group had a control clone with normal TRPV6 expression (Panc-1[Capan-1]-shLuciferase or Panc-1[Capan-1]-mCherry).

The altered expression of TRPV6 channels was assessed in multiple ways. RNA sequencing analysis revealed the expected difference in the respective mRNA levels in Panc-1 cells. Western blot demonstrated that the RNA levels of the stable clones do translate to changes in protein level in Panc-1 cells. TRPV6 levels in Capan-1 cells were assessed via RT-PCR. Moreover, we assessed the impact of altered TRPV6 expression on the Ca^2+^ influx into Panc-1 cells with the Mn^2+^ quench technique. It is based on the principle that Mn^2+^ enters the cells through similar pathways as Ca^2+^ and quenches the fluorescence of the Ca^2+^-sensitive dye Fura2AM. The results are shown in Figure 2.

TRPV6 knockdown impairs the Ca^2+^ influx into Panc-1 cells, as the Mn^2+^-induced decline of Fura2 fluorescence intensity occurs more slowly in Panc-1-shTRPV6 cells than in Panc-1-shLuciferase cells. We found no differences in the quench rate between the Panc-1-mCherry-TRPV6 group and the Panc-1-mCherry cells, suggesting that overexpressed TRPV6 channels are apparently not able to significantly impact the Ca^2+^ influx. 

### 3.3. TRPV6 Knockdown Impairs Proliferation, Cell Survival and Cell Cycle Progression of Panc-1 Cells

The proliferation of Panc-1 cells was investigated as a function of TRPV6 expression by counting the cell numbers at t = 0 h, 24 h and 48 h after seeding. Panc-1-shTRPV6 cells proliferate more slowly than Panc-1-shLuciferase cells (186 ± 2.6% vs. 217 ± 6.5% at t = 48 h) (Figure 3A). On the other hand, Panc-1-mCherry-TRPV6 proliferates faster than the Panc-1-mCherry cells (215 ± 10.4% vs. 155 ± 20.7% at t = 48 h) (Figure 3B). To confirm that these results are not influenced by cell death or cells moving from the ocular field, we counted the mitosis that occurred during the time of the experiment (Figure 3C,D). These results suggest that TRPV6 expression impacts the proliferation of Panc-1 cells positively.

In order to reveal whether the differences in TRPV6 expression impact cell viability, we measured their impact on the intracellular ATP concentration of our Panc-1 cell lines using CellTiter-Glo assay at 24, 48, 72, and 96 h after seeding. The Panc-1-shTRPV6 cells have lower ATP levels than the Panc-1-shLuciferase cells after 48 h (Figure 3E). This indicates that TRPV6 knockdown reduces the cell survival of Panc-1 cells. On the other hand, there is no difference in the ATP levels between the Panc-1-mCherry-TRPV6 group and the Panc-1-mCherry group at any time point (Figure 3F).

These results were also similar in Capan-1 cells. The proliferation levels were dependent on TRPV6 expression. Similarly to Panc-1 cells, in Capan-1 cells, the knockdown of TRPV6 led to lower proliferation rates, as opposed to the overexpression of TRPV6 that generated higher proliferation rates (Figure 4A,B). Furthermore, cell viability was also impacted by TRPV6 expression (Figure 4C,D). 

Cell cycle progression was assessed as a function of TRPV6 channel expression, through propidium iodide staining and flow cytometry, in Panc-1 cells. The focus of the experiment was on the sub-G1 phase of the cell cycle, which represents the fraction of cells with fragmented DNA indicating cell death (Figure 5A). There is a higher percentage of Panc-1-shTRPV6 than Panc-1-shLuciferase cells in the sub-G1 phase (10.8 ± 1.1% vs. 7.8 ± 0.3%) (Figure 5B). This indicates that TRPV6 knockdown induces apoptosis of Panc-1 cells. On the other hand, there is no difference in the sub-G1 phase between Panc-1-mCherry-TRPV6 and Panc-1-mCherry cells (6.5 ± 0.8% vs. 6.6 ± 0.4%) (Figure 5C), indicating that TRPV6 overexpression does not impact the cell cycle progression of Panc-1 cells.

### 3.4. TRPV6 Increases Panc-1 Resistance to Chemotherapeutics

The aggressiveness of tumor cells also implies their resistance to chemotherapeutic drugs. We therefore evaluated the effect of TRPV6 expression on the resistance of Panc-1 cells to gemcitabine, cisplatin and 5-fluorouracil (5-FU). These drugs are commonly used in the treatment of pancreatic cancer. The cells were treated with different concentrations of these agents for 96 h. First, we measured the intracellular ATP concentration of the clones as an indicator of cell viability. We found that TRPV6 knockdown sensitizes Panc-1 cells to 5-FU treatment, as the Panc-1-shTRPV6 cells have lower cell viability levels than the control cells (Panc-1-shLuciferase) after exposure to 1 and 10 µM of 5-FU for 96 h (Figure 6). This suggests that TRPV6 may confer resistance to 5-FU in pancreatic cancer cells. On the other hand, there was no difference in cell viability between the TRPV6 overexpressing cells (Panc-1-mCherry-TRPV6) and the control cells (Panc-1-mCherry) under any condition of 5-FU treatment. There was a tendency for the Panc-1-shTRPV6 cells to have lower cell viability than the Panc-1-shLuciferase group after 96 h of exposure to higher concentrations of gemcitabine or cisplatin. This implies that TRPV6 knockdown might also impact the resistance of Panc-1 cells to gemcitabine or cisplatin.

The next goal was to understand whether the apoptosis of Panc-1 cells induced by chemotherapeutic agents is modulated by the expression level of TRPV6 channels. Therefore, we evaluated annexin V binding to detect the early apoptosis of the cells that were treated with gemcitabine, cisplatin and 5-FU for 24 h. The results are shown in Figure 7. We found that TRPV6 knockdown increases the apoptosis of Panc-1 cells in response to 5-FU and gemcitabine treatment since Panc-1-shTRPV6 cells bind more annexin V than the Panc-1-shLuciferase cells. This confirms that TRPV6 may mediate resistance to 5-FU and, possibly, gemcitabine in pancreatic cancer cells. 

We observed no increase in annexin V binding in TRPV6 overexpressing cells under any of the treatments. Thus, TRPV6 overexpression prevents Panc-1 cells from undergoing apoptosis in response to chemotherapeutics. In contrast, the control cells (Panc-1-mCherry) do bind more annexin V after 5-FU treatment. Collectively, these results suggest that TRPV6 channels contribute to the resistance of pancreatic cancer cells to 5-FU and gemcitabine.

### 3.5. (Collective) Migration of Panc-1 Cells Depends on TRPV6 Channel Expression

To evaluate the effect of TRPV6 expression on the migration of Panc-1 cells, we performed wound-healing assays and monitored the closure of the cell layer during a 20 h period. We found that TRPV6 expression negatively regulates the migration of Panc-1 cells. Hence, TRPV6 knockdown enhances the migration of Panc-1 cells, as Panc-1-shTRPV6 cells close the wound more rapidly than Panc-1-shLuciferase cells (Figure 8A). Conversely, TRPV6 overexpression impairs the migration of Panc-1 cells, as the Panc-1-mCherry-TRPV6 closes the wound more slowly than Panc-1-mCherry cells (Figure 8B).

We further analyzed the dynamics of the collective movement during the wound closure by determining the average distance between four neighboring cells and a central cell during the course of the wound closure. TRPV6 channel knockdown promotes the clustering and a more cohesive movement of Panc-1 cells. The average distance between Panc-1-shTRPV6 cells is much lower than that between the Panc-1-shLuciferase cells (Figure 8C). The average distance between cells is also lower in the Panc-1-mCherry-TRPV6 cells than the control cells (Figure 8D). Nonetheless, this does not imply that TRPV6 overexpression also favors a more cohesive movement of Panc-1 cells. The data gathered indicate that cells overexpressing TRPV6 have nearly half of the wound area closed compared to the control group. This would normally translate to more proximity between the cells, hence dismissing the possibility that the overexpression of the channels is responsible for this effect.

### 3.6. TRPV6 Dysregulation Inhibits Tumor Formation In Vivo

So far, we have seen a correlation between TRPV6 staining and tumor aggressiveness in patient samples which we could, by and large, recapitulate in in vitro experiments. To lend further support to this view, we next examined the effect of TRPV6 expression on tumor growth in vivo in a mouse model. We grafted the four Panc-1 cell lines subcutaneously into nude mice and monitored the tumor size weekly for 10 weeks, either by using a caliper to establish a growth curve within a given time frame or by measuring mCherry intensity in the tumor area (for Panc-1-mCherry or Panc-1-mCherry-TRPV6 cells) (Figure 9A). Surprisingly, we found that both control clones form larger tumors than Panc-1-shTRPV6 and Panc-1-mCherry-TRPV6 cells (Figure 9B,C). The mean tumor volume at 10 weeks is 525 ± 132 mm^3^ for Panc-1-mCherry cells, 410 ± 118 mm^3^ for Panc-1-shLuciferase, 194 ± 30 mm^3^ for Panc-1-shTRPV6 cells and 137 ± 55 mm^3^ for Panc-1-mCherry-TRPV6 cells. These results suggest that TRPV6 expression modulates tumor growth in Panc-1 cells and that both low and high levels of TRPV6 channel expression are strongly inhibitory in our setting. The mice weights did not show any significant differences throughout the experiment (Appendix A). Only two mice were sacrificed due to human endpoints, on the 11th week of experimentation (Appendix A). Both of the mice were from the Panc-1-shLuciferase group.

## 4. Discussion

In this study, we investigated the role of TRPV6 channels in pancreatic ductal adenocarcinoma (PDAC), a highly aggressive and lethal cancer type. We analyzed the expression of TRPV6 channels in tissue samples from 46 patients with PDAC of different stages and found that TRPV6 expression increased with the tumor stage and dedifferentiation status. Our results show that TRPV6 expression is upregulated in PDAC tumor tissues, but only in the later stages and especially in poorly differentiated ones. This suggests that TRPV6 may be involved in the progression and aggressiveness of PDAC. Previous studies have shown that TRPV6 is overexpressed in various cancers, such as prostate, breast, colon, and ovarian cancers [4,5,6]. Regarding PDAC, there are contradictory studies [15,16]. In our cohort, it appears that TRPV6 is prominent in the later, more aggressive stages. It should be stated that the tumor generally metastasizes to neighboring tissues in later stages. Furthermore, poorly differentiated tissues are connected to higher tumor progression [24,25]. Similarly to other tumors, TRPV6 appears to correlate with the aggressiveness of PDAC. Accordingly, the presence of the proliferation marker Ki-67 in patient samples correlates with increased TRPV6 expression. This effect was already demonstrated in prostate cancer tissues [10].

In order to investigate the function of TRPV6 channels in PDAC cell behavior, we generated four stable Panc-1 or Capan-1 cell lines with different levels of TRPV6 expression: a knockdown of TRPV6 expression (Panc-1-shTRPV6 or Capan-1-shTRPV6), an overexpression of TRPV6 (Panc-1-mCherry-TRPV6 or Capan-1-mCherry-TRPV6), and a control clone with normal TRPV6 expression (Panc-1[Capan-1]-shLuciferase or Panc-1[Capan-1]-mCherry). We used these cell lines for in vitro and in vivo experiments to assess the effect of TRPV6 expression on the viability, proliferation, resistance, migration, and tumor growth in nude mice.

These cell lines allowed us to confirm several of the findings in vitro that we made in patient tissue. We found that TRPV6 knockdown reduces the cell proliferation and cell viability in Panc-1 and Capan-1 cells, and the cell cycle progression of Panc-1 cells. Conversely, TRPV6 overexpression enhances proliferation. Capan-1 cells were only used to confirm the in vitro data of cell proliferation and viability, which is one of the major findings in our article. In addition to the proliferation and viability results, TRPV6 knockdown induces apoptosis in Panc-1 cells, as indicated by the increased percentage of cells in the sub-G1 phase. The lack of differences between the overexpression and the control in this experiment can be explained due to 3–5% of the cells being naturally in the sub-G1 phase. These results indicate that TRPV6 is involved in survival and growth of PDAC cells, and that its inhibition may trigger cell death. The mechanisms through which TRPV6 regulates these processes are not fully understood but may involve, at least in part, the modulation of Ca^2+^-dependent signaling pathways. For example, calmodulin, a Ca^2+^-dependent TRPV6 inhibitor, can control the KRas4B pathway that can impact proliferation [26,27]. The inhibition of calmodulin can lead to effects such as lower proliferation levels, inability to progress in the cell cycle or higher cell death [28,29]. Moreover, store-operated Ca2+ channels (SOCE) can act on Akt/mTor or NF-κB pathways modulating proliferation [30]. In prostate cancer, an interaction between SOCE and TRPV6 was already described and could in principle occur in PDAC as well [10]. TRPV6 was shown to increase proliferation, and lead cells to a more aggressive phenotype in prostate cancer cell lines. 

Our results also show that TRPV6 expression influences therapy resistance and the migration of Panc-1 cells, two important features of cancer aggressiveness. We evaluated the effect of TRPV6 expression on the resistance of Panc-1 cells to three chemotherapeutic drugs: gemcitabine, cisplatin, and 5-FU. These drugs are commonly used in the treatment of pancreatic cancer, but often fail to achieve satisfactory outcomes due to the development of drug resistance [31,32,33,34]. We found that TRPV6 knockdown sensitizes Panc-1 cells to 5-FU treatment. On the other hand, we found that TRPV6 overexpression does not affect the sensitivity of Panc-1 cells to 5-FU. However, previous reports are inconclusive in this respect, which is likely due to the fact that chemotherapeutics elicit their cell toxicity by means of different mechanisms. 5-FU makes use of Ca^2+^ as a second messenger to induce the apoptosis of HCT116 colon carcinoma cells [35]. Furthermore, calmodulin inhibition, e.g., through a reduction in the intracellular Ca^2+^ concentration, can impair this process. We therefore believe that our results on chemoresistance of the TRPV6-knockdown cells to 5-FU are unrelated to the Ca^2+^ channel activity of TRPV6 channels. In contrast, our experiments with gemcitabine are in line with the role of TRPV6 as a Ca^2+^ channel. Gemcitabine toxicity is activated by the blockage of calmodulin pathways [36]. The knockdown of TRPV6 can lead to lower levels of calmodulin activation and thereby increase the cytotoxicity of gemcitabine action in PDAC cells [27]. The lack of differences found in the overexpression group when compared to the control, in both chemoresistance and calcium influx, might be an indicator that TRPV6 overexpression does not necessarily translate into higher TRPV6 expression on the membrane. Recently, Kogel et al. indicated that in the HEK293 cell line, TRPV6 has a very short membrane period before a rapid endocytosis [37]. This internalization of TRPV6 might be impacting Ca^2+^ trafficking inside the cells. Thus, TRPV6 could be impacting intracellular homeostasis more than membrane homeostasis. This should be a subject for further studies.

We investigated the effect of TRPV6 expression on the migration of Panc-1 cells using a wound-healing assay. We found that TRPV6 expression negatively regulates the migration of Panc-1 cells. Hence, TRPV6 knockdown enhances the migration of Panc-1 cells, while TRPV6 overexpression impairs it. This effect was quite interesting, seeing that the overexpression of TRPV6 improves proliferation rates, whilst the knockdown gives preference to the migration of the cells. Even though the selection of these phenotypes in the tumor requires different specifications, it was shown that cells can interchange from a more proliferative phenotype to a more migratory one, as was previously suggested in different works [38,39]. Indeed, TRPV6 might rise as a promoter or a switch for these changes. TRPV6 knockdown cells have an increased number of cells in the sub-G1 phase, which clearly demonstrates a deviation from the proliferative phenotype; nonetheless, we could observe that the overexpression of TRPV6 decreases cell motility despite the higher proliferation levels. We also found that TRPV6 knockdown in particular promotes the clustering and a more cohesive movement of Panc-1 cells. In breast and prostate cancer tissues, TRPV6 knockdown appears to impact both cell migration and cell–cell adhesion negatively [12,14].

Our results also show that TRPV6 expression modulates tumor growth in vivo in a mouse model. Surprisingly, we found that both control cell lines form larger tumors than the Panc-1-shTRPV6 and Panc-1-mCherry-TRPV6 cell lines. These results suggest that TRPV6 expression has a biphasic effect on tumor growth in Panc-1 cells and that both low and high levels of TRPV6 can impact tumor development. Future studies will be made on this subject. These findings are unexpected as the in vitro results indicated a positive correlation between TRPV6 expression and cell proliferation. One possible explanation is that TRPV6 expression may affect other aspects of tumor biology, such as angiogenesis, inflammation, or immune response, which are not captured in the in vitro assays [40,41]. Finally, we cannot dismiss the possibility that properties of the tumor microenvironment yet to be identified increase the susceptibility of Panc-1 cells to alterations in their TRPV6 expression.

## 5. Conclusions

Taken together, our results suggest that the TRPV6 channels play a role in PDAC aggressiveness. The inhibition of TRPV6 may reduce the growth and aggressive phenotype of PDAC cells while rendering them more vulnerable to chemotherapy. This could come with the adverse effect of higher motility of the cancer cells. Further studies are needed to elucidate the molecular mechanisms through which TRPV6 regulates PDAC aggressiveness and to evaluate the therapeutic potential of TRPV6 inhibitors in other PDAC models, like pancreatic cancer organoids.

## Figures and Tables

**Figure 1 cancers-15-05769-f001:**
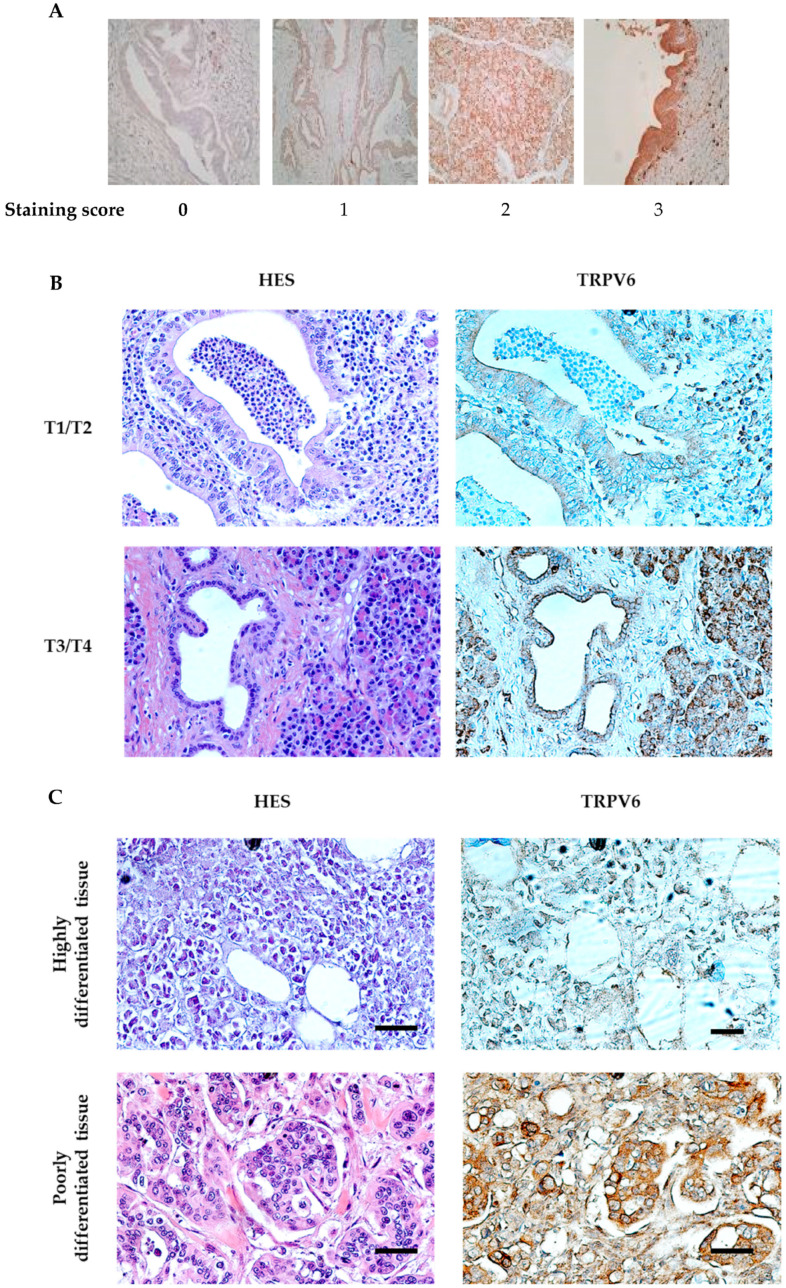
TRPV6 expression correlates with PDAC staging, tissue differentiation and cancer cell proliferation. (**A**) TRPV6 expression was analyzed semi-quantitatively with immunohistochemistry using the TRPV6 antibody Mab82. The expression was scored according to its intensity, ranging from 0 (no staining) to 3 (very intense staining). (**B**) TRPV6 in PDAC tissue from patients in the early stage (top right), late stage (bottom right), with HES staining in the left columns. Scale bars indicate 50 μm. (**C**) TRPV6 staining (brown color) in highly differentiated tissue (top right) or poorly differentiated tissue (bottom right), with HES staining in the left columns. Scale bars indicate 50 μm. (**D**) Ki-67 (blue) and TRPV6 (brown) staining in PDAC tissue. Ki-67 staining (middle column) allowed us to visualize tissue with diverse levels of proliferation. TRPV6 staining (brown color) of the same tissue sections indicates a correlation between TRPV6 expression and the proliferation marker Ki-67, with HES staining in the left columns. Scale bars indicate 50 μm. (**E**) Tumoral and peri-tumoral tissues of 46 PDAC patients were divided into two groups according to their pathological stage (early T1/2 or late T3/4). TRPV6 staining intensities were normalized to the mean values of those found in stage I and II specimen. Error bars represent standard error of the mean of at least three independent tissues. Statistical significance was determined using *t*-test. * *p* < 0.05. (**F**) Tissues were divided into two groups according to their differentiation. Data represent the TRPV6 staining analysis of the two groups. Error bars represent standard error of the mean of at least three patient tissues. Statistical significance was determined using an unpaired *t*-test. * *p* < 0.05. (**G**) TRPV6 staining intensity as a function of the expression of the proliferation marker Ki-67. Error bars represent standard error of the mean of at least three independent tissues. Statistical significance was determined using one-way ANOVA with the Holm–Sidak method. * *p* < 0.05.

**Figure 2 cancers-15-05769-f002:**
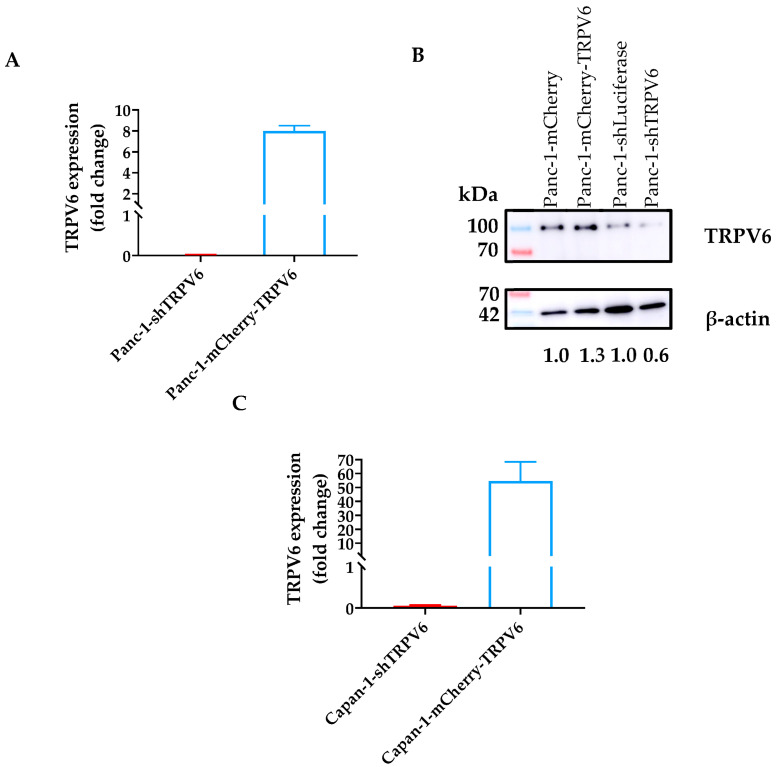
TRPV6 characterization in Panc-1 cells. (**A**) Change in TRPV6 mRNA expression in the knockdown and overexpression Panc-1 clones as determined via RNA sequencing. Changes were normalized to the respective control cell lines and results presented as fold-change. (**B**) Representative immunoblot of whole cell lysates obtained from the stable Panc-1 clones. Channel expression is revealed with a mouse monoclonal anti-TRPV6 antibody. (**C**) Change in TRPV6 mRNA expression in the knockdown and overexpression Capan-1 clones as determined via RT-PCR. (**D**,**E**) Graphical representation of Mn^2+^ quench experiments, evaluated as the fraction of the initial Fura2 fluorescence intensity. (**D**) Panc-1-shLuciferase (black) and Panc-1-shTRPV6 (red); (**E**) Panc-1-mCherry (grey) and Panc-1-mCherry-TRPV6 (blue). (**F**,**G**) Summary of the Mn^2+^ quench experiments. The Mn^2+^ entry rate is calculated as the ratio of the slope of Fura2 fluorescence decline under control conditions and in the presence of Mn^2+^. Error bars represent standard error of the mean of three replicates. Statistical significance was determined using an unpaired *t*-test. * *p* < 0.05.

**Figure 3 cancers-15-05769-f003:**
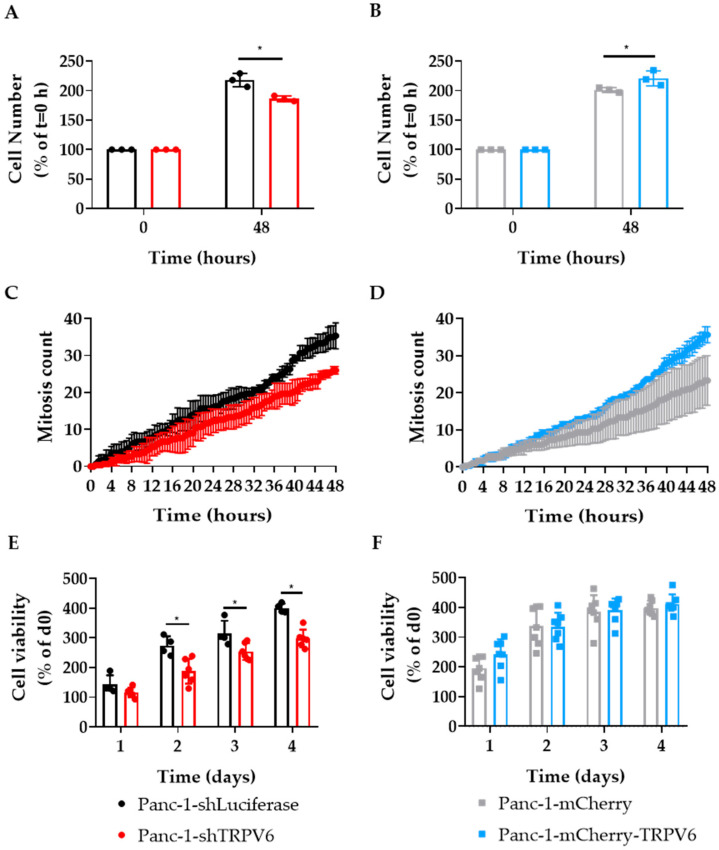
TRPV6 is involved in proliferation and viability of Panc-1 cells. (**A**,**B**) Proliferation of stable Panc-1 cell clones Panc-1-shLuciferase (black) and Panc-1-shTRPV6 (red), Panc-1-mCherry (grey) and Panc-1-mCherry-TRPV6 (blue), 48 h after seeding. Data are expressed as percentage of cell number compared to initial seeding density. Error bars represent standard error of the mean of three replicates. Statistical significance determined using an unpaired *t*-test * *p* < 0.05. (**C**,**D**) Mitosis count Proliferation of stable Panc-1 cell clones Panc-1-shLuciferase (black) and Panc-1-shTRPV6 (red), Panc-1-mCherry (grey) and Panc-1-mCherry-TRPV6 (blue), during 48 h of experiment. (**E**,**F**) Cell viability through ATP concentration of the four stable clones at 24, 48, 72, and 96 h after seeding measured via CellTiter-Glo assay. Data are expressed as percentage of daily luminescence from CellTiter, normalized to the initial luminescence values. Error bars represent standard error of the mean of 4–6 replicates. Statistical significance determined using two-way-ANOVA with Holm–Sidak method * *p* < 0.05.

**Figure 4 cancers-15-05769-f004:**
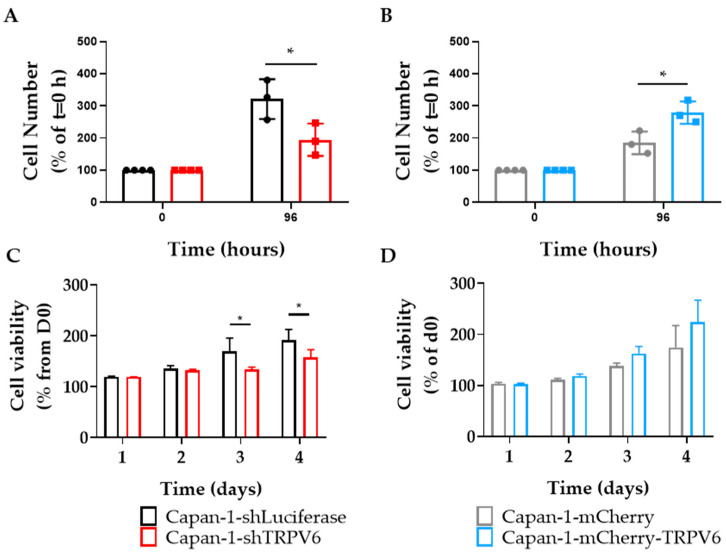
TRPV6 is involved in proliferation and viability of Capan-1 cells. (**A**,**B**) Proliferation of stable Capan-1 cell clones Capan-1-shLuciferase (black) and Capan-1-shTRPV6 (red), Capan-1-mCherry (grey) and Capan-1-mCherry-TRPV6 (blue), 48 h after seeding. Data are expressed as percentage of cell number compared to initial seeding density. Error bars represent standard error of the mean of three replicates. Statistical significance determined using an unpaired *t*-test * *p* < 0.05. (**C**,**D**) Cell viability through MTS of the four stable clones at 24, 48, 72, and 96 h after seeding. Data are expressed as percentage of daily absorbance from MTS reaction, normalized to the initial luminescence values. Error bars represent standard error of the mean of three replicates. Statistical significance determined using unpaired *t*-test * *p* < 0.05.

**Figure 5 cancers-15-05769-f005:**
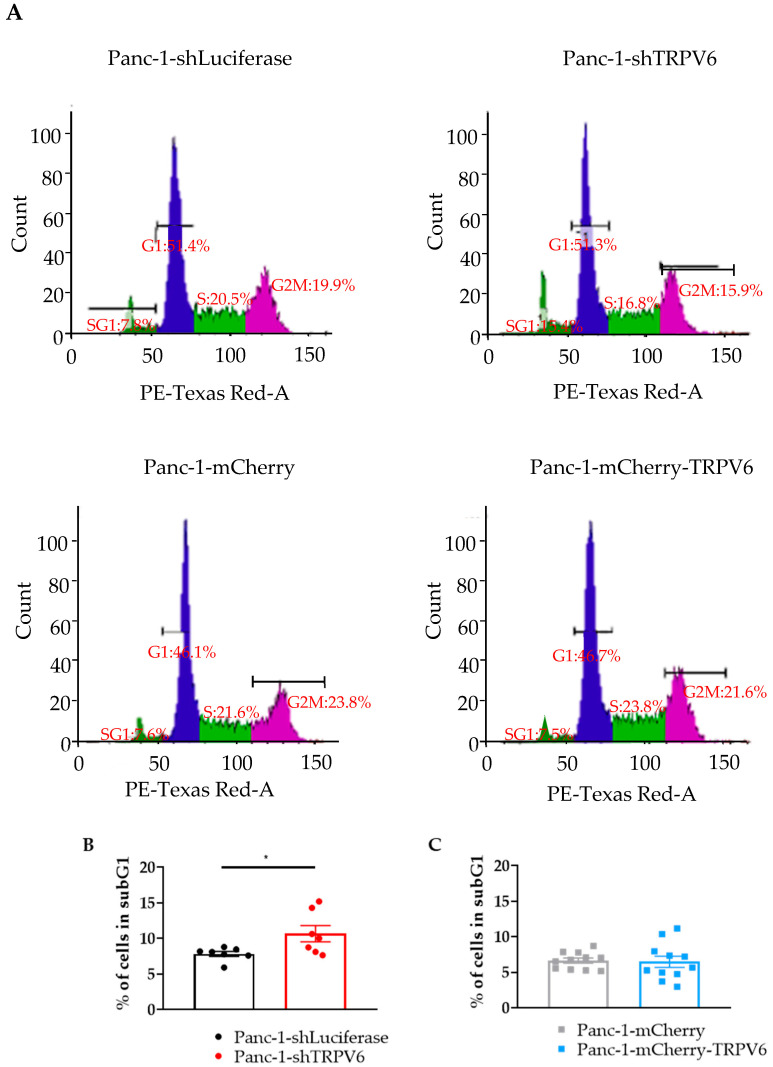
TRPV6 impacts cell cycle progression**.** (**A**) Representative histogram depicting the cell cycle phases of Panc-1-shLuciferase (top left) and Panc-1-shTRPV6 (top right), Panc-1-mCherry (bottom left) and Panc-1-mCherry-TRPV6 cells (bottom right) determined with propidium iodide staining and flow cytometry. Histograms are divided into four groups: SG1 (sub G1), G1, S and G2M (G2 and M phases). *Y* axis indicates the number of cells, whilst *X* axis indicates staining intensity (DNA quantity). (**B**,**C**) Quantification of cells in the sub-G1 phase, which represents apoptotic cells. Error bars represent standard error of the mean of 7–11 replicates. Statistical significance was determined with unpaired *t*-test * *p* < 0.05.

**Figure 6 cancers-15-05769-f006:**
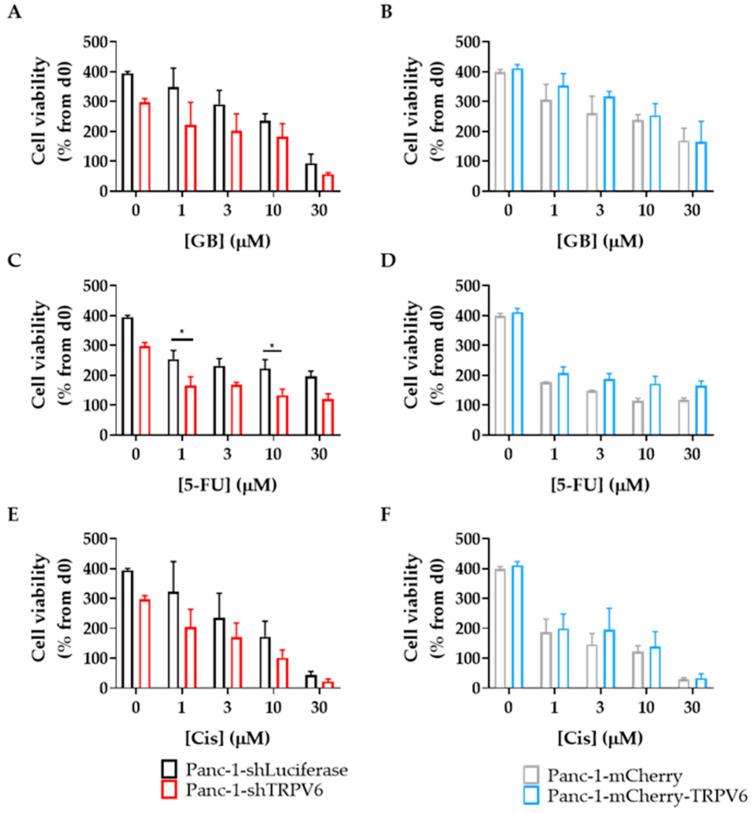
TRPV6 knockdown sensitizes Panc-1 to chemotherapeutics. Cell viability of Panc-1-shLuciferase (black) and Panc-1-shTRPV6 (red), Panc-1-mCherry (grey) and Panc-1-mCherry-TRPV6 cells (blue), after 96 h of treatment with chemotherapeutics, was measured via CellTiter-Glo assay. Stable clones were treated with different concentrations (1, 3, 10 and 30 μM) of (**A**,**B**) gemcitabine (GB), (**C**,**D**) 5-fluorouracil (5-FU) and (**E**,**F**) cisplatin (Cis). Error bars represent standard error of the mean of 4–6 replicates. Statistical significance determined using two-way ANOVA with Holm–Sidak method * *p* < 0.05.

**Figure 7 cancers-15-05769-f007:**
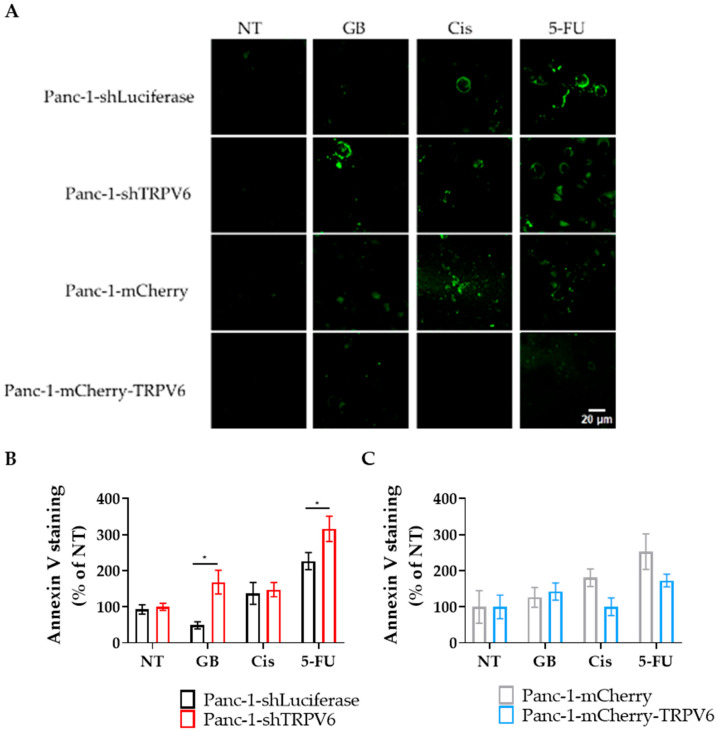
TRPV6 knockdown results in higher Panc-1 cells’ apoptosis in response to chemotherapeutics. Panc-1-shLuciferase (black) and Panc-1-shTRPV6 (red), Panc-1-mCherry (grey) and Panc-1-mCherry-TRPV6 cells (blue) were treated with gemcitabine (GB), cisplatin (Cis) or 5-FU for 24 h. Non-treated cells (NT) served as controls. After that period, an annexin V assay was performed and fluorescence intensity of annexin V staining was measured to determine apoptotic cells. (**A**) Representative images of annexin V staining for the stable clones without treatment (NT) and treated with 3 μM of gemcitabine (GB), cisplatin (Cis) or 5-FU, after 24 h. Scale bar represents 20 μm. (**B**,**C**) Differences in annexin V staining between the TRPV6 knockdown or overexpressing Panc-1 cells and their respective controls. Data are expressed as the percentage of annexin staining intensity, compared to the non-treated group. Each replicate has the mean value of 10 cells per field in seven fields evaluated. Error bars represent standard error of the mean of three replicates. Statistical significance was determined using two-way ANOVA with Holm–Sidak method. * *p* < 0.05.

**Figure 8 cancers-15-05769-f008:**
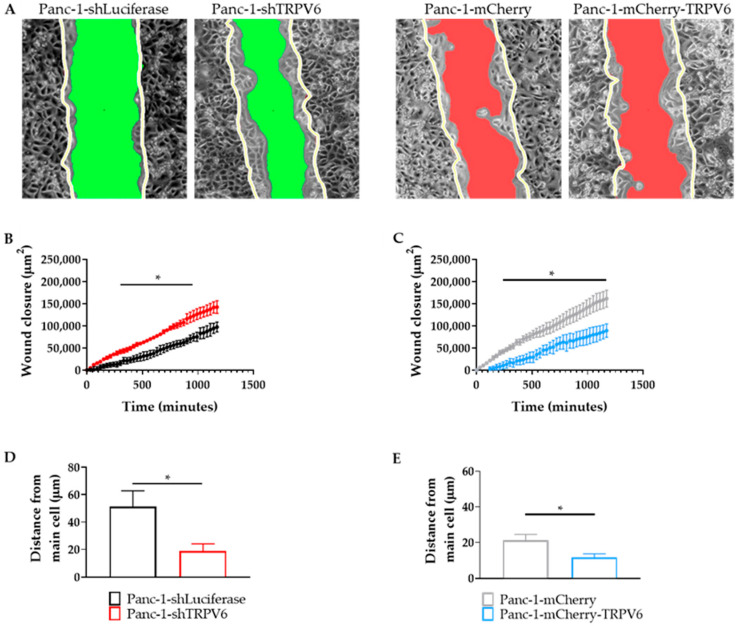
Collective migration of Panc-1 cells is regulated by TRPV6 channel expression. A wound-healing assay was performed and analyzed during a time period of 20 h. (**A**) Representative images of the wound closure of Panc-1-shLuciferase, Panc-1-shTRPV6, Panc-1-mCherry and Panc-1-mCherry-TRPV6 cells. Images depict the differences in area from t = 0 h (represented by the white line) and t = 20 h (represented by either the green (Panc-1-shLuciferase and Panc-1-shTRPV6) or red (Panc-1-mCherry and Panc-1-mCherry-TRPV6) area. (**B**,**C**) Quantification of wound closure in μm^2^ during the course of the experiment. Error bars represent standard error of the mean of four replicates. Statistical significance was determined using multiple *t*-test. * *p* < 0.05. (**D**,**E**) Clustering of cells was analyzed by determining the distance (µm) of four neighboring cells to a “central” cell during the course of the experiment. Results represent the average distance gained throughout the experiments from the neighboring cells to the “central” cell. Error bars represent standard error of the mean of three replicates. Statistical significance was determined using an unpaired *t*-test. * *p* < 0.05.

**Figure 9 cancers-15-05769-f009:**
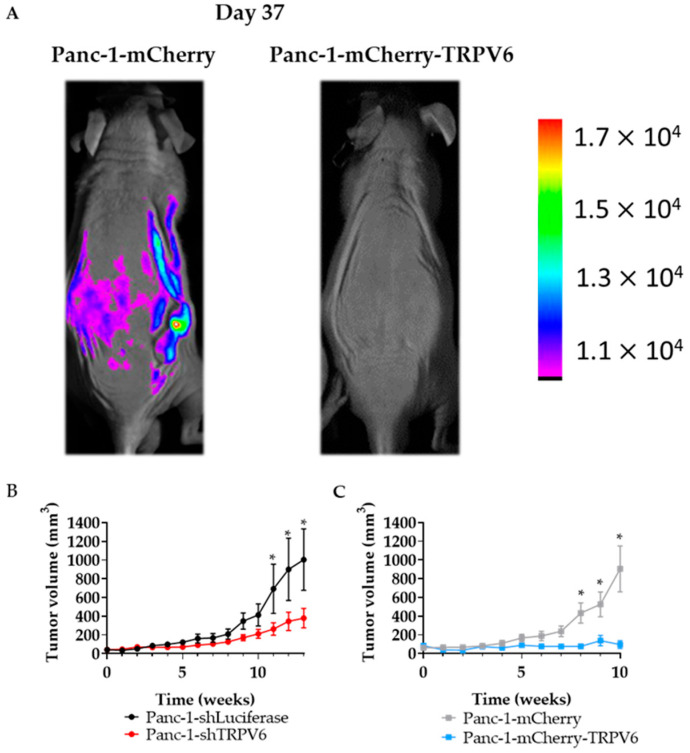
Stable clone with dysregulation of TRPV6 could not promote tumor growth. Nude mice were injected subcutaneously with the four different Panc-1 stable clones. (**A**) Tumor surveillance through mCherry. Representative image comparing the control group (Panc-1-mCherry) with the overexpression of TRPV6 group (Panc-1-mCherry-TRPV6). (**B**) Panc-1-shLuciferase (black) and Panc-1-shTRPV6 (red) (**C**) Panc-1-mCherry (grey) and Panc-1-mCherry-TRPV6 (blue). Tumor growth was evaluated for 10 weeks by measuring and calculating its volume (mm^3^). Error bars represent standard error of the mean of 10 mice. Statistical significance was determined using unpaired *t*-test method between each modulated group and its control. * *p* < 0.05.

## Data Availability

The data presented in this study are available on request from the corresponding author. The data are not publicly available due to no database being created for this work.

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
