# Peer review of "TRPV6 Channel Is Involved in Pancreatic Ductal Adenocarcinoma Aggressiveness and Resistance to Chemotherapeutics"

_cancers, 2023, doi:10.3390/cancers15245769_

Round 1

Reviewer 1 Report (Previous Reviewer 2)

Comments and Suggestions for Authors

In the revised manuscript, the author did a great job and answered all the major and minor concerns with the previous version. The author provided sufficient explanation for major problems wherever applicable. By adding some critical points in the discussion section, the author made significant positive changes in the revised version as per requirement.

I have a minor suggestion to the author in the updated version.

  1. Line 192, in the method section, the author should change the title from ATP quantification to cell viability analysis.
  2. Fig.2 The value has a comma between numbers in the western blot quantification. It is supposed to be a dot (.) instead of a comma. For example, 1,3 and 0,6 should be 1.3 and 0.6, respectively.
  3. Line 352-353, “to confirm these results are not influenced by cell death or cell..” does not seem like a complete sentence.
  4. Fig. 5 legends, in the updated version, should be changed per new figures. Instead of “ATP concentration was measured,” it is supposed to be “cell viability was measured. “ This way reader will not get confused.   

Author Response

Dear reviewer, in the name of the authors, thank you for your kind comments and the text revisions made. Corrections are made point by point below and can be tracked on the manuscript’s text.

Line 192, in the method section, the author should change the title from ATP quantification to cell viability analysis.

The authors accepted the reviewer’s suggestion.

  1. Fig.2 The value has a comma between numbers in the western blot quantification. It is supposed to be a dot (.) instead of a comma. For example, 1,3 and 0,6 should be 1.3 and 0.6, respectively.

The authors accepted the reviewer’s suggestion.

Line 352-353, “to confirm these results are not influenced by cell death or cell..” does not seem like a complete sentence.

We changed the sentence to be more understandable: “To confirm that these results are not influenced by cell death or by cells moving from the ocular field…” lines 372-373

Fig. 6 legends, in the updated version, should be changed per new figures. Instead of “ATP concentration was measured,” it is supposed to be “cell viability was measured. “ This way reader will not get confused.   

The authors accepted the reviewer’s suggestion.

Reviewer 2 Report (New Reviewer)

Comments and Suggestions for Authors

The authors provided comprehensive information on the role of TRPV6 channel in PDAC. However, to have a more complete understanding of the function of this gene, some minor revisions are necessary:

·      To get mor insight into the biological significance of TRPV6 levels, the author must analyze the microenvironment of the tumor generated by sh-Luciferase cells respect to the sh-TRPV6 and compared with the overexpressing mCherry-TRPV6 cells. In particular, Hematoxylin and Eosin staining on tumor sections must be performed.

·      Recently, Delle Cave and colleagues demonstrated that the overexpression of LAMC2 improves the tumorigenic potential of the PDAC cells both in vitro and in vivo (doi: 10.1186/s13046-022-02516-w). The authors must evaluate the expression of LAMC2 (at gene and if possible, also protein level) in sh-Luciferase cells respect to the sh-TRPV6 and compared with the overexpressing mCherry-TRPV6 cells.

Comments on the Quality of English Language

the quality is good

Author Response

Dear reviewer, the authors appreciate your comments and the critics made. Answers will be provided point-by-point.

To get mor insight into the biological significance of TRPV6 levels, the author must analyze the microenvironment of the tumor generated by sh-Luciferase cells respect to the sh-TRPV6 and compared with the overexpressing mCherry-TRPV6 cells. In particular, Hematoxylin and Eosin staining on tumor sections must be performed.

We appreciate the reviewer’s concerns that the TME has also to be investigated, but the focus of our manuscript was to evaluate the role of TRPV6 channels in pancreatic cancer cells. All mechanistic studies were performed with pancreatic cancer cells. It goes beyond the scope of the current manuscript. Nonetheless, we performed an initial experiment on sections obtained from one tumor each of the four different cell lines. This preliminary experiment indicated that CXCR4 and CK18 expression are higher in tumors derived from the control cell lines which would be compatible with their faster tumor growth. 

Recently, Delle Cave and colleagues demonstrated that the overexpression of LAMC2 improves the tumorigenic potential of the PDAC cells both in vitroand in vivo (doi: 10.1186/s13046-022-02516-w). The authors must evaluate the expression of LAMC2 (at gene and if possible, also protein level) in sh-Luciferase cells respect to the sh-TRPV6 and compared with the overexpressing mCherry-TRPV6 cells.

We have evaluated LAMC2 expression on our stable clones, through RNA-seq. As no changes were observed, we decided not to add this information in the manuscript. Below, we have the data indicating the results we found on LAMC2 expression.

This manuscript is a resubmission of an earlier submission. The following is a list of the peer review reports and author responses from that submission.

Round 1

Reviewer 1 Report

Comments and Suggestions for Authors

Gonçalo Mesquita et al. tried to investigate the potential role of TRPV6 playing in drug resistance in single cell line PDAC-1 and FFPE tisssue specimens and nude mice. There are some minor errors to be corrected prior to accepting possibly for publication: 1) Abbreviation, such as TPRV6, 5-FU, usually is inappropriate shwoing in the Title of the paper, 2) Background Introduction should be expanded more to include other groups' work (including but not limited to possible roles of TPRV6 in lung cancer, breast cancer and CRC etc) on other cancer types, as mentioned, the novelty of this study can be thus abundantly clear.

Comments on the Quality of English Language

acceptable but a few errors as mentioned in the "authors"section above.

Author Response

Dear Reviewer, we appreciate your comments. We altered the article title to avoid abbreviations such as PDAC or 5-FU. Nonetheless, we find that having TRPV6 in the abbreviated form in the title can provide for higher manuscript visibility. We made some modifications to the introduction, in order to accommodate a more expanded background for TRPV6 in other groups’ work, following the comments provided by the reviewer. The changes were done in lines 57-67.

"TRPV6 is a Ca2+-permeable channel that belongs to the transient receptor potential (TRP) family of ion channels [4, 5]. It plays an important physiological role in Ca2+ reabsorbing intestinal and renal epithelia. However, TRPV6 is also aberrantly expressed in various cancers, such as prostate, breast, colon, and ovarian cancers [4, 6]. In ovarian cancer, TRPV6 has increased expression in all stages, when compared to normal tissue [7]. Furthermore, inhibition of TRPV6 channels in SKOV-3 xenograft model in mice led to reduced tumor formation. TRPV6 overexpression was also present in breast cancer, where it correlated with invasive areas of the tumor [8]. In prostate cancer cell line LNCaP, TRPV6 inhibition can lead to lower levels of proliferation and increased apoptosis [9]. Translocation of TRPV6 to the plasma membrane and increased aggressiveness was also observed in LNCaP, indicating a clear role for TRPV6 in prostate cancer [10].  Overall, TRPV6 channels promote cancer cell proliferation, survival, migration, and invasion by modulating intracellular Ca2+ signaling and downstream pathways [11-14]."

Reviewer 2 Report

Comments and Suggestions for Authors

In the manuscript, the author studied the crucial role of TRPV6, a calcium-permeable channel, in pancreatic ductal adenocarcinoma (PDAC). The author demonstrated the correlation between TRPV6 expression and PDAC aggressiveness in patient tissue samples. To understand the critical role of TRPV6, the author utilized the knockdown and overexpression approach and performed several cellular end-point assays, including cell proliferation, cell viability, and cell apoptosis. The manuscript is well written; however, essential points must be considered.

Major Points:-

1.       In the manuscript, the crucial role of TRPV6 has been demonstrated using knockdown and overexpression approach. The observed phenotype in knockdown cells is not reversed by TRPV6 overexpression when compared to relative control, except for cell number (Fig. 3A, 3B) and wound closure (Fig. 6B and 6C). Still, the author has misappropriated written, “conversely TRPV6 overexpression yielding opposing effect- line no.22”; “line no. 41-43” and line no. 489-490.

This needs to be clarified. Furthermore, the author should discuss why overexpression of TRPV6 does not impact the tested end-point assay compared to the control cells. This clarification should be included in the discussion session.

2.       Knockdown or overexpression was verified by RNA sequence analysis. Notably, the change in TRPV6 expression should be confirmed at the protein level. In Fig. 2A, the bar graph shows the TRPV6 expression change in Panc1-shTRPV6 compared to Panc1-Shluciferase. The value is supposed to be negative for TRPV6 knockdown cells, but in the bar graph, it shows around zero.  The author should revisit this data.  

3.       The author claims TRPV6 impacts cell proliferation by performing a cell counting assay. However, Fig. 4a shows that knockdown of TRPV6 has more cells in the sub-G0 phase, an apoptosis marker. Therefore, it might be possible the observed lower cell number in Panc1-Sh-TRPV6 (Fig. 3a) may be the outcome of apoptosis induction rather than reduced cell proliferation. The author should perform other cell proliferation assays, such as time kinetics cell proliferation (0,24,48,96h) or dye-dilution methods to demonstrate the impact of TRPV6 on cell proliferation.

4.       Fig 6, the author observed more wound closure in TRPV6 knockdown PDAC cells. In addition, the TRPV6 knockdown also observed less value for distance from primary cells. On the other hand, overexpression of TRPV6 has less wound closure. However, the distance from primary cells (as a marker for collective cell movement) is also less in TRPV6 overexpressed cells. How? The author should describe this in the result section.

5.       Fig 5, it seems the knockdown of TRPV6, a red bar (at day 0), has a reduced value compared to the control bar. In this circumstance, it is hard to correlate the impact of chemotherapeutics between two cell lines. The author should normalize each data point to day 0 for the respective line. Then, the % cell viability should be calculated and represented for each issue.

It’s hard to believe in the current data representation, but it seems like TRPV6 impacts 5-FU drug resistance, but gemcitabine and cisplatin have minimal effects. However, the author overstated the title in the result section and the abstract session. It needs to be considered.

Minor points:-

1.       Fig.1 A, the images are poor quality, particularly for 0,1 and 2 staining core images. 1B and 1C, it is highly recommended to provide a good-resolution image. In the current image, the TRPV6 staining intensity counting is hard.

2.       In Annexin V staining assay, the author has plotted the value by counting ten cells per visual field. The author should provide information like how many numbers of areas they used for cell count. In general, 50-100 cells should be counted for each field, and a minimum of three fields should be considered for each condition.

3.        There is much diversity in data representation throughout the paper. For example, the font size for the y-axis or x-axis differs between the figure or in the same figure. The author should use one format and one font size to represent data. In the bar graph, some have individual data plots, whereas others have a mean and standard error of the mean. It is advisable to stick to one format, and I think the author should represent each bar graph with an individual data plot.

4.       The author measured cell viability using cell titer glow. For this data, the author represented data in ATP concentration. It is better to represent in % cell viability instead of ATP concentration.

5.       On several occasions, the applied statistical test for significance needs to be revisited. For example, in Fig. 3C, 3D, and Fig5, the author should use an unpaired t-test instead of a 2-way-ANOVA for statistical analysis.  

6.       In Fig 4A, the author should provide the value for the % cells present in each cell-cycle phase. In addition, in Fig.4A, the X-axis and Y-axis titles are not easy to read.  

7.       In the result section(3.6), line no 451-453, “mean ± …..” values are missing.

Author Response

Dear Reviewer, we are thankful for the thorough revision of our manuscript and for the points taken.

Major Points:-

  1. In the manuscript, the crucial role of TRPV6 has been demonstrated using knockdown and overexpression approach. The observed phenotype in knockdown cells is not reversed by TRPV6 overexpression when compared to relative control, except for cell number (Fig. 3A, 3B) and wound closure (Fig. 6B and 6C). Still, the author has misappropriated written, “conversely TRPV6 overexpression yielding opposing effect- line no.22”; “line no. 41-43” and line no. 489-490.

This needs to be clarified. Furthermore, the author should discuss why overexpression of TRPV6 does not impact the tested end-point assay compared to the control cells. This clarification should be included in the discussion session.

The reviewer has pointed out an incongruence in the text, in which we wrongly indicated that TRPV6 overexpression reversed phenotypic reactions in cell viability and cell apoptosis. This is already corrected in the lines 41 and 42. Furthermore, we hope to have explained the reasons why overexpression of TRPV6 does not impact some tested end-point assay compared to the control cells. This can be seen in lines 520-522 and in 551-558. The text modifications are the following:

“Conversely, TRPV6 overexpression yielded opposing outcomes in terms of proliferation and migration of Panc-1 cells.”

“We found that TRPV6 knockdown reduces the cell proliferation, cell viability, and cell cycle progression of Panc-1 cells. Conversely, TRPV6 overexpression enhances proliferation. In addition, TRPV6 knockdown induces apoptosis in Panc-1 cells, as indicated by the increased percentage of cells in the sub-G1 phase. The lack of differences between the overexpression and the control in this experiment can be explained due to 3-5% of the cells being naturally in the sub-G1 phase.”

“The lack of differences found in the overexpression group when compared to the control, in both chemoresistance and calcium influx, might be an indicative that TRPV6 overexpression does not necessarily translate into higher TRPV6 expression on the membrane. Recently, Kogel et al. indicated that in HEK293 cell line TRPV6 has a very short membrane period before a rapid endocytosis [37]. This internalization of TRPV6 might be impacting Ca2+ trafficking inside the cells. Thus, TRPV6 could be impacting intracellular homeostasis more than membrane homeostasis. This should be a subject for further studies.”

  1. Knockdown or overexpression was verified by RNA sequence analysis. Notably, the change in TRPV6 expression should be confirmed at the protein level. In Fig. 2A, the bar graph shows the TRPV6 expression change in Panc1-shTRPV6 compared to Panc1-Shluciferase. The value is supposed to be negative for TRPV6 knockdown cells, but in the bar graph, it shows around zero.  The author should revisit this data.  

Unfortunately, there was a mistake in the graph presented in the version of the manuscript presented to the reviewer. This has been corrected. The data refers to the fold change between the stable clones and respective controls. Fig1 and lines 311-313. Western blot was performed to confirm TRPV6 expression in the stable clones. A new image was added to figure 2.

“Change of TRPV6 mRNA expression in the knockdown and overexpression clones as determined by RNA sequencing. Changes were normalized to the respective control cell lines and results presented as fold-change.”

“The altered expression of TRPV6 channels was assessed in multiple ways. RNA sequencing analysis revealed the expected difference of the respective mRNA levels. Western blot demonstrated that the RNA levels of the stable clones do translate to changes in protein level.”

  1. The author claims TRPV6 impacts cell proliferation by performing a cell counting assay. However, Fig. 4a shows that knockdown of TRPV6 has more cells in the sub-G0 phase, an apoptosis marker. Therefore, it might be possible the observed lower cell number in Panc1-Sh-TRPV6 (Fig. 3a) may be the outcome of apoptosis induction rather than reduced cell proliferation. The author should perform other cell proliferation assays, such as time kinetics cell proliferation (0,24,48,96h) or dye-dilution methods to demonstrate the impact of TRPV6 on cell proliferation.

The material and methods part was reviewed to better explain how the cell count was made, in lines 168-171. Furthermore, a new graph was included to figure 3. This might increase the perception that TRPV6 is in fact impacting proliferation.

“Images of proliferating Panc-1 cells were acquired every other 30 min for 48 h, starting 2 hours after plating. Cell proliferation was also assessed through mitosis count. This allowed us to exclude cells migrating from other areas.”

  1. Fig 6, the author observed more wound closure in TRPV6 knockdown PDAC cells. In addition, the TRPV6 knockdown also observed less value for distance from primary cells. On the other hand, overexpression of TRPV6 has less wound closure. However, the distance from primary cells (as a marker for collective cell movement) is also less in TRPV6 overexpressed cells. How? The author should describe this in the result section.

After analysing the reviewer comments we improved the results section for this part, so a better comprehension of these data can be made, in lines 445-453.

“The average distance between Panc-1-shTRPV6 cells is much lower than between the Panc-1-shLuciferase cells (Fig.7C). The average distance between cells is also lower in the Panc-1-mCherry-TRPV6 cells than the control cells (Fig.6D). Nonetheless, this does not imply that TRPV6 overexpression also favors a more cohesive movement of Panc-1 cells. The data gathered indicates that cells overexpressing TRPV6 have nearly half of wound area closed than the control group. This would normally translate to more proximity between the cells, hence dismissing the possibility that the overexpression of the channels is responsible for this effect.”

  1. Fig 5, it seems the knockdown of TRPV6, a red bar (at day 0), has a reduced value compared to the control bar. In this circumstance, it is hard to correlate the impact of chemotherapeutics between two cell lines. The author should normalize each data point to day 0 for the respective line. Then, the % cell viability should be calculated and represented for each issue.

It’s hard to believe in the current data representation, but it seems like TRPV6 impacts 5-FU drug resistance, but gemcitabine and cisplatin have minimal effects. However, the author overstated the title in the result section and the abstract session. It needs to be considered.

In figure 5 the data represents cell viability, depending on the concentration of a determined compound, after 96 hours. The data is already normalized to day 0 and the concentration 0 corresponds to the basal data presented in figure 3E and 3F, day 4. The impact of TRPV6 should not be understated here. 5-FU treatment clearly affects the knockdown cells in both viability assay and apoptosis assay. Although the other compounds don’t produce such blunt results, there is a tendency for the knockdown cells to have lower viability levels when treated with gemcitabine and cisplatin. In the annexin V assay, gemcitabine produces significant results, indicating a higher sensitivity of the cells to this compounds. The same can be seen with the overexpression group, where there is a tendency for these cells to have higher levels of viability and lower levels of apoptosis with the treatment of cisplatin and 5-fluorouracil.

Minor points:-

  1. Fig.1 A, the images are poor quality, particularly for 0,1 and 2 staining core images. 1B and 1C, it is highly recommended to provide a good-resolution image. In the current image, the TRPV6 staining intensity counting is hard.

Included new images with better resolution. Fig1

  1. In Annexin V staining assay, the author has plotted the value by counting ten cells per visual field. The author should provide information like how many numbers of areas they used for cell count. In general, 50-100 cells should be counted for each field, and a minimum of three fields should be considered for each condition.

More information on the analysis of this data was added in material and methods lines 202 and 203; and in the figure’s legend lines 428 and 429.

“Cell death was analyzed by intensity of fluorescence in 10 cells on each area. 7 areas were counted for each experiment.”

“Data are expressed as the percentage of annexin staining intensity, compared to the non-treated group. Each replicate has the mean value of 10 cells per field in 7 fields evaluated. Error bars represent standard error of the mean of 3 replicates.”

  1. There is much diversity in data representation throughout the paper. For example, the font sizefor the y-axis or x-axis differs between the figure or in the same figure. The author should use one format and one font size to represent data. In the bar graph, some have individual data plots, whereas others have a mean and standard error of the mean. It is advisable to stick to one format, and I think the author should represent each bar graph with an individual data plot.

We made changes in order to harmonize the graphs throughout the manuscript.

  1. The author measured cell viability using cell titer glow. For this data, the author represented data in ATP concentration. It is better to represent in % cell viability instead of ATP concentration.

We accepted the reviewer’s comments.

  1. On several occasions, the applied statistical test for significance needs to be revisited. For example, in Fig. 3C, 3D, and Fig5, the author should use an unpaired t-test instead of a 2-way-ANOVA for statistical analysis.  

We do believe that the statistical tests done in the figures indicated by the reviewer are in fact well chosen. This comes from the fact that we are evaluating 2 different factors in these graphs (TRPV6 expression and time) (TRPV6 expression and concentration of the compound). We believe 2-way anova is the strongest statistical test we can use in these specific cases.

  1. In Fig 4A, the author should provide the value for the % cells present in each cell-cycle phase. In addition, in Fig.4A, the X-axis and Y-axis titles are not easy to read.  

Added the percentage results of each phase for the representative histograms and improved the axis and legend. Fig.4A

  1. In the result section(3.6), line no 451-453, “mean ± …..” values are missing.

We apologize for this mistake. The values were added to the text in the line 481 and 482.

Reviewer 3 Report

Comments and Suggestions for Authors

The manuscript "TRPV6 is involved in PDAC aggressiveness and chemo-2 resistance to 5-FU and gemcitabine" by Mesquita et. al. studies the role of TRPV6 in drug resistance of PDAC cells to 5-FU and gemcitabine. The results are interesting and overall a good study. I have a few concerns below:

1. TRPV6 protein expression must be shown by western blot in Panc-1-shTRPV6 cells.

2. The authors must use a chemical inhibitor of TRPV6 like econazole and ruthenium red (PMID: 34725357) and check whether Panc-1 cells are better sensitized to 5-FU and gemcitabine. 

3. The authors must use a second PDAC cell line like Capan-2 etc. to replicate their studies. Data from Panc-1 cells alone should not be considered.

I recommend the manuscript for a major revision. Thank you.

Author Response

Dear Reviewer, we appreciate the comments made to our manuscript. Nonetheless, due to the time period given for the review it would be impossible for us to accomplish all the requirements made.

1. TRPV6 protein expression must be shown by western blot in Panc-1-shTRPV6 cells.

Western blot was performed to confirm TRPV6 expression in the stable clones. A new image was added to figure 2.

“The altered expression of TRPV6 channels was assessed in multiple ways. RNA sequencing analysis revealed the expected difference of the respective mRNA levels. Western blot demonstrated that the RNA levels of the stable clones do translate to changes in protein level.”

  1. The authors must use a chemical inhibitor of TRPV6 like econazole and ruthenium red (PMID: 34725357) and check whether Panc-1 cells are better sensitized to 5-FU and gemcitabine. 

Although it would be interesting to chemically inhibit TRPV6, such compounds are not available. Those mentioned by the reviewer, econazole and ruthenium red are not specific to TRPV6 and could promote changes that would wrongly imply a role for the channels. Econazole can inhibit other TRP channels like TRPM2 and more importantly TRPV5 (PMID: 29323279). This channel could be of interest due to its proximity with TRPV6 and could be a player for cells to compensate for TRPV6 knockdown. As for ruthenium red, the impact is even bigger than econazole. Ruthenium red can block most of the TRP channels of the vanilloid family. Our main scope of this manuscript is solely on the function of TRPV6.

  1. The authors must use a second PDAC cell line like Capan-2 etc. to replicate their studies. Data from Panc-1 cells alone should not be considered.

Although we understand the concerns of the reviewer, we believe that our manuscript provides a good basis for the understanding of TRPV6 role in PDAC, with the use of human tissues, in vivo and in vitro assays. We have also tried to use Capan-1 in the beggining of the experiments. Due to the slower speed of proliferation and clustering like formation of these cells, we ended up abandoning it, to focus on all the other experiments with Panc-1. Furthermore, with the time given for this review it would be impossible to satisfy this request.

Round 2

Reviewer 3 Report

Comments and Suggestions for Authors

Dear Authors,

I agree that the chemical inhibitors might not be commercially available for you to try out in your studies, but including a second PDAC cell line is absolutely required. You must replicate your studies in another PDAC cell line to confirm your conclusions. I hereby reject this manuscript in its present form and encourage you to expand your studies with another PDAC cell line of your choice and then submit the manuscript as a new submission.

Thank you